# Biological Characteristics of *Verticillium dahliae MAT1-1* and *MAT1-2* Strains

**DOI:** 10.3390/ijms22137148

**Published:** 2021-07-01

**Authors:** Lin Liu, Ya-Duo Zhang, Dan-Dan Zhang, Yuan-Yuan Zhang, Dan Wang, Jian Song, Jian Zhang, Ran Li, Zhi-Qiang Kong, Steven J. Klosterman, Xiao-Feng Dai, Krishna V. Subbarao, Jun Zhao, Jie-Yin Chen

**Affiliations:** 1College of Horticulture and Plant Protection, Inner Mongolia Agricultural University, Hohhot 010018, China; A_lin95@163.com (L.L.); y2015k@126.com (Y.-Y.Z.); zhj19890128@126.com (J.Z.); 2Team of Crop Verticillium wilt, c/o State Key Laboratory for Biology of Plant Diseases and Insect Pests, Institute of Plant Protection, Chinese Academy of Agricultural Sciences, Beijing 100193, China; zyduo8@126.com (Y.-D.Z.); zhangdandan@caas.cn (D.-D.Z.); wangdan_star@163.com (D.W.); songjian_01@126.com (J.S.); liran01@caas.cn (R.L.); kongzhiqiang@caas.cn (Z.-Q.K.); daixiaofeng_caas@126.com (X.-F.D.); 3United States Department of Agriculture, Agricultural Research Service, Crop Improvement and Protection Research Unit, Salinas, CA 93905, USA; Steve.Klosterman@usda.gov; 4Department of Plant Pathology, University of California, Davis, c/o U.S. Agricultural Research Station, Salinas, CA 93905, USA

**Keywords:** *Verticillium dahliae*, *MAT1-1* and *MAT1-2* idiomorphs, Growth characteristics, Pathogenicity

## Abstract

*Verticillium dahliae* is a soil-borne plant pathogenic fungus that causes Verticillium wilt on hundreds of dicotyledonous plant species. *V. dahliae* is considered an asexually (clonal) reproducing fungus, although both mating type idiomorphs (*MAT1-1* and *MAT1-2*) are present, and is heterothallic. Most of the available information on *V. dahliae* strains, including their biology, pathology, and genomics comes from studies on isolates with the *MAT1-2* idiomorph, and thus little information is available on the *MAT1-1 V. dahliae* strains in the literature. We therefore evaluated the growth responses of *MAT1-1* and *MAT1-2* *V. dahliae* strains to various stimuli. Growth rates and melanin production in response to increased temperature, alkaline pH, light, and H_2_O_2_ stress were higher in the *MAT1-2* strains than in the *MAT1-1* strains. In addition, the *MAT1-2* strains showed an enhanced ability to degrade complex polysaccharides, especially starch, pectin, and cellulose. Furthermore, several *MAT1-2* strains from both potato and sunflower showed increased virulence on their original hosts, relative to their *MAT1-1* counterparts. Thus, compared to *MAT1-1* strains, *MAT1-2* strains derive their potentially greater fitness from an increased capacity to adapt to their environment and exhibit higher virulence. These competitive advantages might explain the current abundance of *MAT1-2* strains relative to *MAT1-1* strains in the agricultural and sylvicultural ecosystems, and this study provides the baseline information on the two mating idiomorphs to study sexual reproduction in *V. dahliae* under natural and laboratory conditions.

## 1. Introduction

Sexual reproduction, a ubiquitous feature of the eukaryotic kingdom, can accelerate the adaptation to continuously changing ecological niches. It also allows the repair of random epigenetic or conventional genetic damage by recombination with homologous chromosomes [1,2]. However, approximately 20% of the fungal species reproduce only by asexual means, with no recognized sexual cycle [3,4]; and in Ascomycotina, nearly 40% of the taxa were deemed asexual [5]. Such strictly asexual organisms are thought to be less flexible than sexual organisms, relying solely on random mutations to adapt to changing environments; therefore, they are often considered evolutionary dead ends, mainly due to the absence of meiotic recombination, resulting in increased accumulation of deleterious mutations, an effect known as the Muller’s ratchet [6,7,8,9]. Especially for plant pathogens that are engaged in an endless arms race with their hosts, attempting to ward off invading pathogens, relatively quick adaptations to coevolve with the host immune system are advantageous for evolutionary success [10]. However, accumulating evidence over the past few decades suggests that many of these supposedly asexual fungi might actually have the potential for sexual reproduction [11,12]. Examples include *Candida albicans*, *Aspergillus*
*flavus*, *Aspergillus fumigatus*, and *Aspergillus parasiticus* [13,14,15,16,17], as well as the biotechnologically relevant species *Penicillium chrysogenum*, *Penicillium roqueforti*, and *Trichoderma reesei* [18,19,20,21].

In most fungi, the mating-type locus represents a relatively small region of the genome, less than a few thousand base pairs, encoding transcription factors that act as master regulators of sexual reproduction [22]. One idiomorph of the mating-type locus contains a critical gene that encodes an α domain (*MAT1-1-1*), while the other contains a gene that encodes a DNA-binding domain of the high-mobility group (HMG) (*MAT1-2-1*) [23]. In heterothallic species, sexual recombination occurs only between partners of opposite mating types; and in contrast, homothallic species normally contain both MAT loci, which can either be located at a single MAT locus or on different chromosomes [12,24]. However, the regulatory impact of *MAT*-encoded transcription factors and *MAT* gene expression alone does not necessarily indicate sexual potential. In asexual fungi, *MAT* genes likely possess other important functions beyond mating. In the asexual *Aspergillus oryzae*, *MAT1-1* and *MAT1-2* specifically regulate over 1000 genes, including many with unknown functions [25]. In *Penicillium chrysogenum*, in addition to being involved in sexual development, *MAT*-controlled processes include asexual development, pallet morphology, polar hyphal growth, conidiospore germination, and secondary metabolite synthesis such as penicillin [26].

*V. dahliae* is a soil-borne, broad host-range plant pathogen that invades the xylem of susceptible plant species to cause vascular wilts [27,28]. Hundreds of dicotyledonous plants are hosts of *V. dahliae*, including many economically important crops such as lettuce, cotton, strawberries, and tomato [27,29,30,31]. *V. dahliae* is generally considered solely asexual (clonal), because the sexual cycle has never been documented either in nature or in the laboratory. The mating type idiomorph distribution in nature is skewed overwhelmingly toward *MAT1-2*, and extensive chromosomal rearrangements among different *V. dahliae* genomes could potentially interfere with meiosis. Moreover, the population structure of global *V. dahliae* strains is highly clonal [32,33,34,35]. This clonal structure is further complicated by abundant population subdivisions, including six main vegetative compatibility groups (VCGs) [36,37,38], races 1 (R1) and 2 (R2) that respond to host resistance genes in tomato and lettuce [39,40], a race 3 (R3) defined only in tomato [41], and two pathotypes, defoliating (D) and non-defoliating (ND) based on the presence or absence of defoliation caused by individual strains [42]. Some evidence supports a well-conserved ancestral or cryptic sexual reproduction in *V. dahliae*, including the two mating types in *V. dahliae*, and constitutively expressed sex-related genes [35]. Therefore, even though the evidence in the literature supports the clonal expansion of *V. dahliae*, the preservation of the machinery for sexual reproduction and the cryptic and ancestral sexual cycle in *V. dahliae* suggest that the potential for sexual reproduction exists in the fungus.

Most of the available information on *V. dahliae* strains, including the biology, pathology, genetics, and genomics, is derived from studies on strains carrying the *MAT1-2* idiomorph [43,44,45], and information on the *MAT1-1* idiomorph is scarce. The lack of information on the *MAT1-1* idiomorph has limited our understanding and research into the sexual mode of reproduction in *V. dahliae*. In recent years, *V. dahliae* strains with the *MAT1-1* idiomorph have been frequently recovered from potato and sunflower in the Inner Mongolia Autonomous Region of China, and the distribution of the two mating-type strains is close to 1:1, which will significantly increase the temporal and spatial proximity of two *V. dahliae* mating-type strains to potentially enable mating and increase the risk of sexual recombination. A more efficient genome evolution that results in a novel progeny may ensue. Therefore, a comparative analysis of the basic growth and pathogenic characteristics of strains carrying the two mating-type idiomorphs will provide important data for the genetic analysis of the two idiomorphs and to evaluate the possibility of sexual reproduction in *V. dahliae*.

## 2. Results

### 2.1. Characteristics of V. dahliae MAT1-1 Strain P48 and MAT1-2 Strain P50

To obtain comparative information on the *MAT1-1* and *MAT1-2* strains, *V. dahliae* strain P48 with *MAT1-1* and P50 with *MAT1-2* isolated from potato in Inner Mongolia, China, were selected. The upper sides of the P48 and P50 cultures displayed white dense mycelia on the PDA medium that progressively darkened on the undersurface of the cultures as the melanized microsclerotia were produced, at about 7 days after incubation (Figure 1A). During this interval, strain P50 produced more melanin than P48 (Figure 1A). The conidia from both strains were hyaline, elongated in clusters on phialides borne in whorls on branched aerial hyphae (Figure 1A), typical characteristics of *V. dahliae* [46].

A number of PCR primers have been designed to amplify loci or markers associated with race (D or ND phenotype) and mating type idiomorphs [35,46,47,48]. The PCR assays indicated that strain P48 had the *MAT1-1* marker and P50 the *MAT1-2* marker, confirming that P48 and P50 represented strains with *MAT1-1* and *MAT1-2* idiomorphs, respectively (Figure 1B). Strain P50 was also characterized as the ND pathotype and belonged to race 2, while strain P48 contained the race 2 marker, and it neither carried the D nor the ND marker, suggesting that both P48 and P50 were race 2 strains and that P50 likely is not able to cause the defoliating phenotype on hosts such as olive, cotton, and okra. The well-characterized, highly virulent defoliating strain Vd991 represents the D pathotype, belongs to race 2, and carries the *MAT1-2* marker. It was used as a control in these experiments.

To detect the sequence conservation and variability of the two mating-type master regulatory genes in strains P48 and P50, the coding sequences of the *MAT1-1-1* and *MAT1-2-1* genes from these two strains were aligned with the corresponding genes from eight other sexually reproducing fungi, two of which were homothallic (*Fusarium graminearum*, *Aspergillus nidulans*) and six were heterothallic (*Magnaporthe grisea*, *Aspergillus fumigatus*, *Neurospora crassa*, *Saccharomyces cerevisiae*, *Penicillium marneffei*), along with those of the *V. dahliae* strain VdLs.17, which has been used as the reference genome in many studies [43]. The results showed that both P48 and P50 had the conserved α or HMG domains similar to other sexual fungi, and most amino acids in the α or HMG domains were also highly conserved (Figure 1C).

### 2.2. Growth of MAT1-1 and MAT1-2 Strains under Different Culture Conditions

The growth characteristics of the *MAT1-1* strain P48 were compared with those of the *MAT1-2* strain P50 in response to a range of temperatures, pH treatments, light/dark regimes, carbon sources, and stress conditions. Furthermore, the growth of five additional isolates of *MAT1-1* (S2, S11, S109, P51, and P56) and *MAT1-2* (S1, S12, S23, P52, and P90) strains from sunflower (S) and potato (P) was also determined to ensure that the results obtained with P48 and P50 were not unique to these strains. Detailed information on these strains is provided in Appendix A.

#### 2.2.1. Temperature

*V. dahliae* has an optimal temperature range of 22−27 °C, and limited growth occurs above 32 °C [49]. The *MAT1-1* strain P48 produced a predominantly white mycelium, but the *MAT1-2* strain P50 produced less mycelium, displayed faster colony growth, and accumulated higher amounts of melanin following incubation for 15 days at 25 °C (Figure 2A).

The growth rates of both P50 and P48 were significantly lower at 28 °C and 30 °C relative to 25 °C (Figure 2A). However, even at 28 °C, the colony diameter of the *MAT1-2* strain P50 was significantly larger than that of the *MAT1-1* strain P48 (Figure 2A,B). At 30 °C, the growth of both strains was restricted and was not significantly different (Figure 2A,B). These results suggest that temperatures <28 °C had fewer influences on the *MAT1-2* strain P50 than on the *MAT1-1* strain P48, but once the temperature exceeded 30 °C, the growth of both strains was nearly arrested.

Additional *MAT1-1* and *MAT1-2 V. dahliae* strains, under the optimum temperature (25 °C), showed no detectable differences in their growth rates for up to 15 days incubation, except that the *MAT1-1* strains mainly developed white mycelia with less melanin accumulation than the *MAT1-2* strains (Figure 2C and Appendix A). At 28 °C and 30 °C, the diameters of the *MAT1-2* strains were significantly higher than those of the *MAT1-1* strains (Figure 2C and Appendix A). Higher temperatures therefore restrict growth and melanin production in both *MAT1-1* and *MAT1-2* strains, but these effects were more pronounced in the *MAT1-1* strains than in the *MAT1-2* strains.

#### 2.2.2. pH

Verticillium wilt of cotton normally occurs in near-neutral to alkaline soils at a pH in the range 6–9. At pH 5.5 or below, the growth, microsclerotia production, and survival of *V. dahliae* are generally inhibited [49]. Under laboratory conditions, the optimum pH for *V. dahliae* on the PDA medium was approximately 6.5. To determine the differences in the growth of the two idiomorphs, three PDA plates at pH 7.0, 8.0, and 9.0 each were incubated at 25 °C. Strain P50 produced more melanin under these pH values than at pH 6.5 (Figure 2A and Figure 3A). Melanin accumulation increased with increasing pH, with the maximum melanin enrichment occurring at pH 9.0 (Figure 3A). However, the melanin production by the *MAT1-1* strain P48 showed no obvious change with the pH (Figure 3A). Increasing pH did not alter the growth rates of either strain, but at each pH tested, P50 exhibited a higher colony growth than strain P48 (Figure 3A,B). Alkaline pH stimulated the accumulation of melanin by the two mating-type strains, but had no quantifiable influence on colony growth rates.

The growth rates of the five additional *MAT1-1* strains were not altered by the pH, but melanin accumulation was reduced at pH 8 and 9, regardless of whether they were isolated from potato or sunflower. In contrast, *MAT1-2* strains from potato and sunflower displayed higher growth rates at higher pH, and melanin production was relatively unaffected compared to the *MAT1-1* strains (Figure 3C and Appendix A). Overall, relative to the *MAT1-1* strains, *MAT1-2* strains not only grew faster but also accumulated more melanin under near-neutral and alkaline pH.

#### 2.2.3. Light/Dark

The growth of strains P48 and P50 was evaluated on a PDA medium at 25 °C under dark and light conditions. Under the dark conditions examined, both the P48 and P50 strains produced melanin, although the growth rates and melanin accumulation in strain P50 were higher relative to those of P48 (Figure 4A). However, under the light conditions examined, neither P48 nor P50 produced visible melanin, with both forming only white mycelia (Figure 4A). The colony diameter of P48 was smaller under light conditions than under dark conditions, but the colony diameters of the P50 strain showed no obvious differences between light and dark conditions (Figure 4B). Under the light or dark conditions, the growth rates of P50 were higher than those of P48 (Figure 4B). These results suggested that light interfered with the melanin production in both strains and limited the rate of growth of the *MAT1-1* strain P48 but did not influence the growth rate of the *MAT1-2* strain P50.

The influence of light was further examined on the growth of five additional strains from *MAT1-1* and *MAT1-2*. Under the light conditions tested, there was no visible melanin production in the *MAT1-1* strains, with the exception of S2 (Appendix A). Similarly, melanin accumulation in the *MAT1-2* strains was also minimal under light, with only strains S23 and P52 producing small amounts of melanin (Appendix A). The growth rate of *MAT1-2* strains whether under dark or light conditions was higher relative to that of the *MAT1-1* strains; and *MAT1-2* strains produced more melanin under the dark conditions than the *MAT1-1* strains (Figure 4C and Appendix A). These results indicate that light restricted melanin production in both mating-types but had relatively less influence on the growth rate of *MAT1-2* strains versus *MAT1-1* strains.

#### 2.2.4. Carbon Source

*Verticillium* species are capable of metabolizing a wide range of carbon sources, including glucose, mannose, rhamnose, sucrose, xylose, and cellobiose [50,51,52]. To examine the complex nutrient utilization capacity of the two *MAT* strains, their growth rates and phenotypes were evaluated on media containing different carbon sources, including the relatively simple disaccharide sucrose, but also complex polysaccharides like starch, pectin, and cellulose. The growth rates of the *MAT1-2* strain P50 were higher compared to that of the *MAT1-1* strain P48 on each of the media containing different carbon sources, and P50 also produced more melanin (Figure 5A,B). The higher growth of strain P50 suggested that the *MAT1-2* strain has an increased capacity to degrade complex carbon sources, especially the complex polysaccharides pectin, cellulose, and starch. The *MAT1-1* strain P48 produced mainly white mycelia on starch-, sucrose– and pectin-containing media with no apparent melanin production (Figure 5A). Melanin production was also significantly enriched in the *MAT1-2* strain P50 on starch, sucrose, and pectin media, in addition to producing the typical white mycelia (Figure 5A). The growth of both mating type strains was limited on the medium containing cellulose as a carbon source, and there was little visible melanin production on this medium (Figure 5A). Both strains utilized pectin, but their ability to utilize cellulose was relatively weak compared with other complex polysaccharides examined.

Based on these results, three complex carbon sources were further evaluated on the larger panel of *MAT1-1* and *MAT1-2* strains. The *MAT1-2* strains produced more melanin and less white mycelia than the *MAT1-1* strains on the starch medium (Appendix A). When pectin served as the carbon source, both *MAT1-1* and *MAT1-2* strains produced melanin and mycelia (Figure 5C and Appendix A). When cellulose was the carbon source, similar to P48 and P50, the growth of other *MAT1-1* and *MAT1-2* strains was limited, with scant mycelia, small colony diameters, and little melanin accumulation (Appendix A). The colony diameters of all *MAT1-2* strains were larger relative to those of the *MAT1-1* strains on media containing the three selected carbon sources (Figure 5B and Appendix A). Since pectin and cellulose are some of the chief constituent complex polysaccharides in plants, *MAT1-2* strains may have an enhanced capability, compared to *MAT1-1* strains, to degrade plant cell wall components during infection and colonization processes.

### 2.3. Stress Tolerance (Osmotic Stress, Oxidative Stress, Cell Wall Integrity Stress)

To evaluate the stress tolerance of the two mating type strains, their growth on media that imposed osmotic stress (sorbitol-containing medium at three concentrations: 1, 1.5, and 2 mol/L), oxidative stress (H_2_O_2_-containing medium at three concentrations: 0.9, 1.2, and 1.8 mmol/L), and cell wall integrity stress (Congo red-containing medium at three concentrations: 100, 200, and 300 μg/mL) was evaluated.

Under osmotic stress, the growth rate of both P48 and P50 decreased as the sorbitol concentration increased, and there was no apparent melanin production under sorbitol stress. While there were no differences in the growth rates between the two strains under sorbitol concentrations of 1 and 1.5 mol/L, the growth rate of the P50 strain was higher relative to that of strain P48 at 2 mol/L sorbitol (Appendix A). Similarly, under cell wall integrity stress, the growth rates of both P48 and P50 decreased as the Congo red concentration increased (Appendix A). Strain P50 produced more melanin at 100 and 200 μg/L Congo red than P48 (Appendix A). Under the cell wall stress conditions tested, there were also no apparent differences in growth rate between the two strains (Appendix A). The growth rates of both strains decreased as the oxidative stress from H_2_O_2_ concentration increased. However, there was almost no growth of strain P48 at the concentration of 1.8 mmol/L H_2_O_2_ (Figure 6A,B). Unlike the other stress conditions examined, the growth rate of the *MAT1-2* strain P50 was higher than that of the *MAT1-1* strain P48 in response to H_2_O_2_, indicating that the *MAT1-2* strain P50 had a greater resistance to oxidative stress than the *MAT1-1* strain P48.

Colony diameters of the additional *MAT1-1* strains evaluated were significantly smaller in size as the H_2_O_2_ concentration increased. At 1.8 mmol/L H_2_O_2_ none of the *MAT1-1* strains grew except P56, which produced very small colonies (Appendix A). The growth rates of the *MAT1-2* strains were also slower as the H_2_O_2_ concentrations increased, the only difference being that the colony diameters were larger relative to those observed in the *MAT1-1* strains

### 2.4. Pathogenicity of MAT1-1 and MAT1-2 Strains on Their Original Hosts

The virulence of the *MAT1-1* and *MAT1-2* strains was evaluated on potato or sunflower. Most of the *MAT1-2* strains from potato were more virulent than the *MAT1-1* strains on potato, except for strain P57 (Figure 7). However, the overall virulence of *MAT1-2* and *MAT1-1* strains from sunflower was not significantly different on sunflower (Figure 7).

## 3. Discussion

We compared the growth and the virulence of *V. dahliae* strains of opposite mating types derived from sunflower and potato. The *MAT1-2* strains exhibited higher growth rates and melanin production under high temperatures, alkaline pH, light, and oxidative stress (Figure 2, Figure 3 and Figure 4 and 6). Additionally, *MAT1-2* strains showed an enhanced ability to degrade complex polysaccharides, especially starch, pectin, and cellulose (Figure 5). Correspondingly, some *MAT1-2* strains from potato also exhibited increased virulence relative to the *MAT1-1* strains on potato. However, the difference in virulence between the two mating types were not apparent on sunflower (Figure 7). This research documents the overall higher adaptability of the *MAT1-2* strains, that confers them greater fitness. This may help explain the major imbalance in the frequencies of the two mating idiomorphs in the agricultural and sylvicultural ecosystems, and provides information for future inquiries into the potential sexual cycle of *V. dahliae* under experimental conditions.

Temperature, as the single most important environmental factor, not only governs the germination of *V*. *dahliae* microsclerotia and disease development but also influences the geographic distribution of *V. nonalfalfae* and *V. dahliae* [49,53]. *V. dahliae* is mainly distributed in subtropical and temperate areas and has an optimal temperature in the range 22–27 °C, but limited growth occurs above 32 °C. Devaux and Sackston [54] demonstrated that almost no microsclerotia formed at temperatures over 30 °C. Abundant microsclerotia formed between 18 and 30 °C within 2−5 days and after 30 days at 5 °C, and no microsclerotia formed at 32 °C [55]. Colony growth and melanin production were altered as temperatures rose from 25 °C to 30 °C. At 30 °C, especially, the colony growth rates decreased and there was almost no melanin accumulation (Figure 2A,B). Similarly, the larger panel of *MAT1-1* and *MAT1-2* strains showed no growth rate differences at 25 °C. The growth rate of the *MAT1-2* strains was higher relative to those of the *MAT1-1* strains as the temperature increased (Figure 2C), and also accumulated more melanin up to 30 °C (Appendix A). High temperature affected *MAT1-2* strains less than *MAT1-1* strains, indicating a stronger temperature tolerance of *V. dahliae MAT1-2* strains. Melanin production in *V. dahliae* is also linked with protection against high temperature stress [56].

The maintenance of a constant pH was the single most important factor in maintaining optimal growth rates of *V. dahliae*, and in cotton production, the disease normally occurs in soils that are neutral to alkaline (pH 6–9). Under acidic pH (5.5 or lower), the mycelial growth, microsclerotia production, and survival are all reduced [49]. Increases in the pH slightly reduced growth and melanin synthesis in the *MAT1-1* strain P48 (Figure 3A). In contrast, growth was not influenced significantly with increasing pH, but melanin production increased in the *MAT1-2* strain P50 (Figure 3A). Overall, the optimum pH for *MAT1-1* strains was approximately 7.0, but the *MAT1-2* strains could grow and produce melanin under a broad pH range.

For a soil-borne and plant xylem-invading fungus like *V. dahliae*, a dark environment may represent an optimal condition for *V. dahliae* growth and proliferation. Some of the earliest research findings on melanin production in *V. albo-atrum* (reclassified as *V. alfalfae* and *V. nonalfalfae* by Inderbitzin et al. [46]) suggest that its resting mycelium formed sooner in total darkness than in daylight [57]. Caroselli et al. [58] also claimed that “green light” inhibited growth and that more microsclerotia were formed in “red light” and darkness. Conidial production by *V. dahliae* was greatly enhanced, but microsclerotia production was completely inhibited by blue light [59,60]. Some previous studies indicated that melanin occurs as electron-dense granules in the outer cell wall of microsclerotia, and melanin deposition is typically coupled with later stages of microsclerotial formation in *V. dahliae* [61,62]. Under light conditions, except for a few strains, there was almost no melanin production for all strains as compared to incubation in the dark (Figure 4A and Appendix A). Under light conditions, reduced melanin synthesis decreases the survival of *V. dahliae*, as these strains are highly sensitive to UV radiation [56]. This also corresponds to previous research showing that unmelanized conidia are more prone to rapid desiccation and damage by UV radiation [58,63].

*Verticillium* spp. are capable of metabolizing a wide range of carbon sources, including glucose, fructose, arabinose, galactose, mannose, cellobiose, ribose, and xylose, as well as complex polysaccharides such as starch, pectin, and cellulose [52]. *V. dahliae* encodes a wide range of secreted pectinolytic and cellulolytic enzymes that may initiate infection and symptom development [43,64]. Thus, the capacity of *V. dahliae* to utilize different carbon sources is crucial for the infection of various host plants and the proliferation in host cells, including in the xylem, which consists of complex polysaccharides such as pectin, cellulose, and lignified cellulose [65]. The growth rates and melanin production of the *MAT1-2* strain P50 were higher than those of the *MAT1-1* strain P48 when sucrose, starch, cellulose, and pectin were used as carbon (Figure 5 and Appendix A). The cellulose utilization by both strains, however, was lower compared with the other three carbon sources. This was consistent with the growth of other *V. dahliae* strains on a cellulose-containing medium [66]. Although the growth rates of *MAT1-2* strains were higher than in the *MAT1-1* strains on media containing different polysaccharides, the melanin production in the *MAT1-2* strains in general was not limited, unlike in strain P48 (Figure 5C and Appendix A). Nonetheless, the *MAT1-2* strains generally had a higher polysaccharide utilization ability relative to the *MAT1-1* strains.

Congo red, sorbitol, and H_2_O_2_ have been widely used to detect fungal cell wall integrity and resistance to high concentrations of osmotic regulators and oxidative stress, respectively. Congo red interacts with various polysaccharides and exhibits a particularly high affinity with chitin and cellulose, which disturbs fungal cell wall morphogenesis [67,68,69]. The osmotic regulator sorbitol at high concentrations causes osmotic stress [70,71]. In response to pathogen infection, plants tend to produce more H_2_O_2_ to inhibit pathogen proliferation, and the reactive oxygen scavenging capacity has been viewed as an important evaluation index to assess pathogen infection ability and pathogenicity [56]. Under the Congo red- and sorbitol-induced stress, the growth rates and melanin production of both *MAT1-1* strain P48 and *MAT1-2* strain P50 were impaired, except for the higher melanin production at relatively low Congo red concentrations (100 and 200 μg/mL) by strain P50 (Appendix A). These results suggest that the sensitivities of the two strains to Congo red and sorbitol were similar. However, *MAT1-1* and *MAT1-2* strains responded differently to H_2_O_2_ stress, suggesting differing reactive oxygen scavenging abilities in the two mating-type populations.

In conclusion, under a higher temperature, alkaline pH, response to light, and oxidative stress, the *MAT1-2* strains exhibited higher growth rates and melanin production relative to those of the *MAT1-1* strains. In addition, the *MAT1-2* strains also had a higher complex polysaccharide degradation ability, suggesting that the *MAT1-2* strains had wider environmental adaptability, that gave them greater fitness and survivability. These advantages may also confer the *MAT1-2* strains higher virulence on select host plants. As demonstrated in this study, there was higher virulence of the *MAT1-2* strains from potato when assayed on potato but such differences were not apparent with the two mating-type strains from sunflower when assayed for virulence on sunflower. Nonetheless, relative to the *MAT1-1* strains, *MAT1-2* strains exhibit an increased ability for environmental and potential host adaptation. The results of this research support the hypothesis that *MAT1-2* strains are more adaptable, and therefore far more abundant in nature than the *MAT1-1* strains. This research also provides the baseline information on the two mating idiomorphs to pursue studies of a potential sexual cycle in *V. dahliae* under different experimental conditions.

## 4. Materials and Methods

### 4.1. Growth of V. dahliae Strains under Various Environmental Factors

To determine the growth rates and colony morphology of *V. dahliae MAT1-1* and *MAT1-2* strains under different temperatures, pH changes, carbon sources, light and dark conditions, and conditions causing osmotic and cell wall stress, conidial suspensions of every strain were prepared at 5 × 10^6^ conidia/mL, and 2 μL was transferred to the center of Petri plates containing different media.

For temperature sensitivity assays, *MAT1-1* and *MAT1-2* strains were incubated on a potato dextrose agar (PDA) medium at 25 °C, 28 °C, and 30 °C, respectively. Three plates for each temperature were incubated in each experiment, and the experiment was repeated three times. The pH of the PDA medium was adjusted to 7.0, 8.0, and 9.0 using 1 M HCl or NaOH to determine the pH sensitivity of all isolates at 25 °C. Growth under light and dark conditions was tested on a PDA medium, with each strain incubated at 25 °C. For the carbon source utilization assays, the strains were cultured at 25 °C in basic C’zapek medium agar plates that included sucrose (30.0 g/L), starch (17.0 g/L), pectin (10.0 g/L), or cellulose (10.0 g/L) as the carbon source, respectively [72]. Again, three plates at each pH and isolate were incubated per experiment, and the experiment was repeated three times.

PDA plates containing 0.9, 1.2, and 1.8 mmol/L 30% H_2_O_2_ were used to test all *MAT1-1* and *MAT1-2* isolates to oxidative stress. For osmotic stress experiments, strains P48 and P50 were inoculated and cultured on a complete medium (CM, Yeast extract 6.0 g/L, Casein acids hydrolysate 6.0 g/L, sucrose 10.0 g/L) at 25 °C supplemented with 1, 1.5, and 2 mol/L sorbitol, respectively. For cell wall stress assays, strains P48 and P50 were incubated on CM plates that were supplemented with 100, 200, and 300 μg/mL Congo red (Sigma-Aldrich) at 25 °C.

The growth of all strains under different culture conditions were evaluated after 15 days of incubation by measuring the colony diameters. Each strain was cultured on five plates for each of the treatments, and the experiments were repeated two times. Unpaired Student’s *t*-tests were performed to determine the statistical significance among strains at *p* ≤ 0.05, *p* ≤ 0.01, or *p* ≤ 0.005.

### 4.2. DNA Extraction and Genotype Identification Assays of V. dahliae Strains

High conidial suspensions (1 × 10^8^ conidia/mL) of *V. dahliae* strains P48, P50, and Vd991 were placed on a PDA medium and incubated at 25 °C, and the mycelium was collected after 7 days. The total genomic DNA was extracted for the identification of mating types, races, and D and ND types. Genomic DNA of each isolate was extracted using a FastPure Plant DNA Isolation Mini Kit (Vazyme, Nanjing, China) following the manufacturer’s instructions and was stored at −20 °C for polymerase chain reaction (PCR) assays.

For the genotype identification assays, D/ND, race1/2, and *MAT1-1*/*MAT1-2* were determined by PCR with previously developed primers [35,73,74] (Appendix A). All PCR assays in this study were performed in 20 μL reaction volumes using 2×Taq Mester Mix (Dye Plus) P112-AA (Vazyme, Nan Jing, China). PCR was performed under the following conditions: an initial 94 °C denaturation step for 10 min, followed by 30 cycles at 94 °C for 30 s, annealing at 58 °C for 30 s, extension at 72 °C for 1 min, and a final extension of 10 min at 72 °C. The PCR products were detected by 1% agarose (Sigma-Aldrich, St. Louis, MO, USA) gel dyed with GelStain (TransGen, Beijing, China) and electrophoresis for 20 min at 120 V in 1×TAE buffer. Then the image was obtained with the Bio-Rad’s ChemiDoc XRS system.

### 4.3. Gene Cloning and Bioinformatics Analysis

The mycelia of strains P48 and P50 were collected 7 days after culture at 25 °C on a PDA medium. The total RNA of P48 or P50 was extracted using the Total RNA Miniprep Kit (Axygen, MA, USA), and first-strand cDNA was synthesized using the RevertAid First Strand cDNA Synthesis Kit (Thermo Scientific, Waltham, MA, USA). The targeted genes (*MAT1-1-1* and *MAT1-2-1*) were amplified from the prepared cDNA using the primers listed in Appendix A.

Protein sequence alignments deduced from *MAT1-1-1* and *MAT1-2-1* of strains P48 and P50, respectively, were performed with other published fungal mating-type protein sequences (*Verticillium dahliae* VdLs.17: BAG83052.1 and BAG12301.1, *Fusarium graminearum*: AAG42809.1 and AAG42810.1, *Magnaporthe grisea*: BAC65083.1 and BAC65094.1, *Penicillium marneffei*: ABC68484.1 and ABC68485.1, *Aspergillus fumigatus*: AAX83123.1 and XP_754989.2, *Aspergillus nidulans*: EAA63189.1 and XP_659566.1, *Neurospora crassa*: AAC37478.1 and P36981.2, and *Saccharomyces cerevisiae*: EDN62161.1 and NP_010790.3). A sequence alignment analysis and domain prediction were conducted using Clustal X2 (http://www.clustal.org/clustal2/ (accessed on 24 August 2020)) and BoxShade (https://embnet.vital-it.ch/software/BOX_form.html (accessed on 24 August 2020)).

### 4.4. Virulence of MAT1-1 and MAT1-2 Strains

Ten *MAT1-1* (P48, P51, P56, P89, P91, S2, S11, S109, S31, S29) and *MAT1-2* (P50, P52, P57, P88, P90, S1, S12, S23, S47, S38) *V. dahliae* strains were selected for virulence assays. Assays were performed as described previously by Xiao et al. [75], with some modifications. All strains were cultured on a PDA medium for 7 days, transferred to a wheat bran medium, and incubated at 25 °C for 15 days until the medium was covered with mycelium. Conidia were harvested with distilled water and filtered with Miracloth (Solarbio, Beijing, China), and a 1 × 10^7^ conidia/mL inoculum suspension was used in the virulence assays.

Sunflower (cv. He 15) seeds were sown with 5 seeds in each pot, and potatoes (cv. LD 5009) were planted with two tubers in each pot, with each tuber containing at least 2–3 bud eyes. The pots were incubated at 28 °C in the greenhouse with alternating 16 h/8 h light- dark periods. Four-week-old sunflower and potato seedlings were inoculated by flooding 200 mL conidial suspension as described by Alkher et al. and Xiao et al. [75,76]. The plants were examined for symptoms weekly, and the quantification of disease parameters was performed at three weeks post-inoculation following the methods of Xiao et al. and Alkher et al. [75,76]. Five pots of sunflowers and ten pots of potatoes were used for each *V. dahliae* strain, and the experiment was repeated three times. The quantification of disease parameters was done as follows: Potato Verticillium wilt was rated on a 0 to 5 scale, where 0 = healthy plants with no foliar chlorosis and necrosis; 1 = visible leaf chlorosis with less than 1% necrosis; 2 = up to 40% chlorosis and 1–20% necrosis; 3 = up to 65% chlorosis and 21–35% necrosis; 4 = 100% chlorosis, 36–70% necrosis; and 5 = 100% chlorosis, 71–100% necrosis.

Verticillium wilt on sunflower was rated on a 0–4 scale, where 0 = healthy plants with no foliar chlorosis; 1 = 25% foliar chlorosis and stunting; 2 = 26–50% foliar chlorosis and stunting; 3 = 51–75% foliar chlorosis and stunting; 4 = more than 75% of the leaves with severe chlorosis and stunting, along with plants’ death.

The disease indices for the potato and sunflower virulence assays were calculated according to the formula: =100 × ∑ (number of diseased leaves in each scale rating × representative value in each rating)/(total number of leaves examined × maximum rating value).

## Figures and Tables

**Figure 1 ijms-22-07148-f001:**
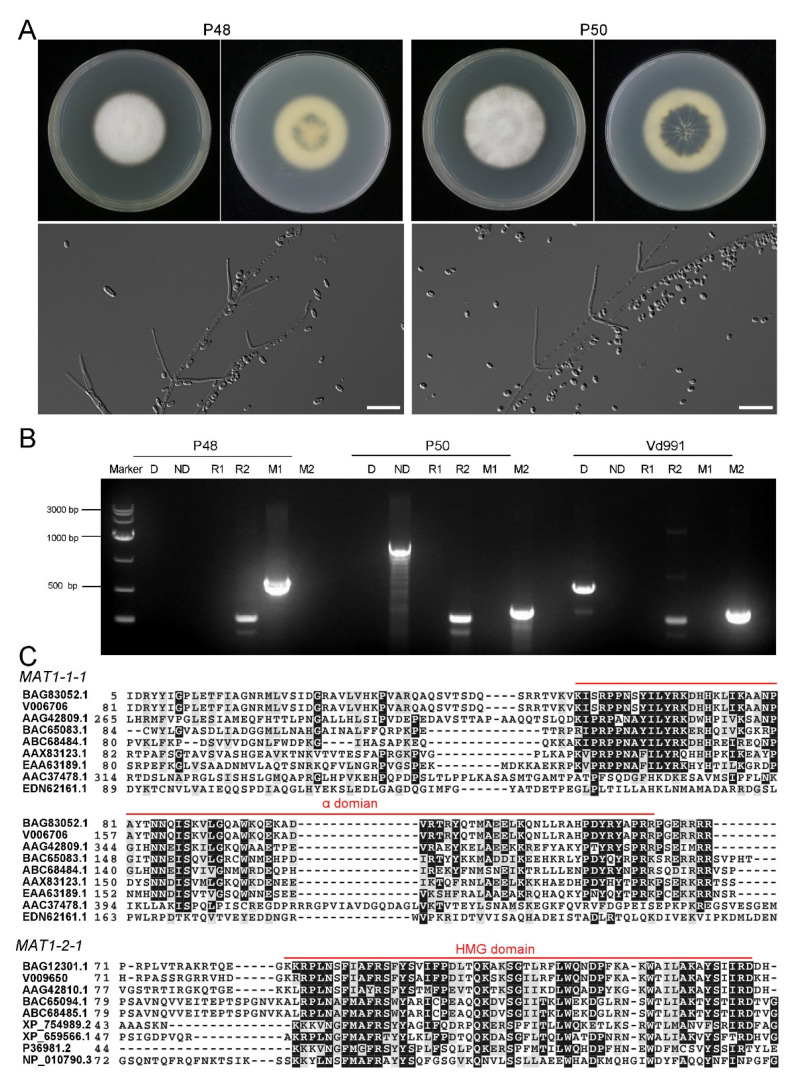
Morphology and genetic characterization of the *Verticillium dahliae* strains P48 and P50. (**A**) Colony phenotype and conidial morphology of *V. dahliae* stains P48 and P50 on PDA medium. (**B**) Results of molecular identification of *Verticillium dahliae* strains P48 and P50 using primers specific for *MAT1-1* (M1)/*MAT1-2* (M2), race 1 (R1)/race 2 (R2), and defoliating (D)/non-defoliating (ND) characteristics. (**C**) α and HMG domain comparison of sex-related transcriptional regulation factors *MAT1-1-1* and *MAT1-2-1* in P48 and P50 with other sexual fungi. BAG83052.1 and BAG12301.1: *Verticillium dahliae* VdLs.17; V006706 and V009650: *Verticillium dahliae* P48; AAG42809.1 and AAG42810.1: *Fusarium graminearum*; BAC65083.1 and BAC65094.1: *Magnaporthe grisea*; ABC68484.1 and ABC68485.1: *Penicillium marneffei*; AAX83123.1 and XP_754989.2: *Aspergillus fumigatus*; EAA63189.1 and XP_659566.1: *Aspergillus nidulans*; AAC37478.1 and P36981.2: *Neurospora crassa*; EDN62161.1 and NP_010790.3: *Saccharomyces cerevisiae*.

**Figure 2 ijms-22-07148-f002:**
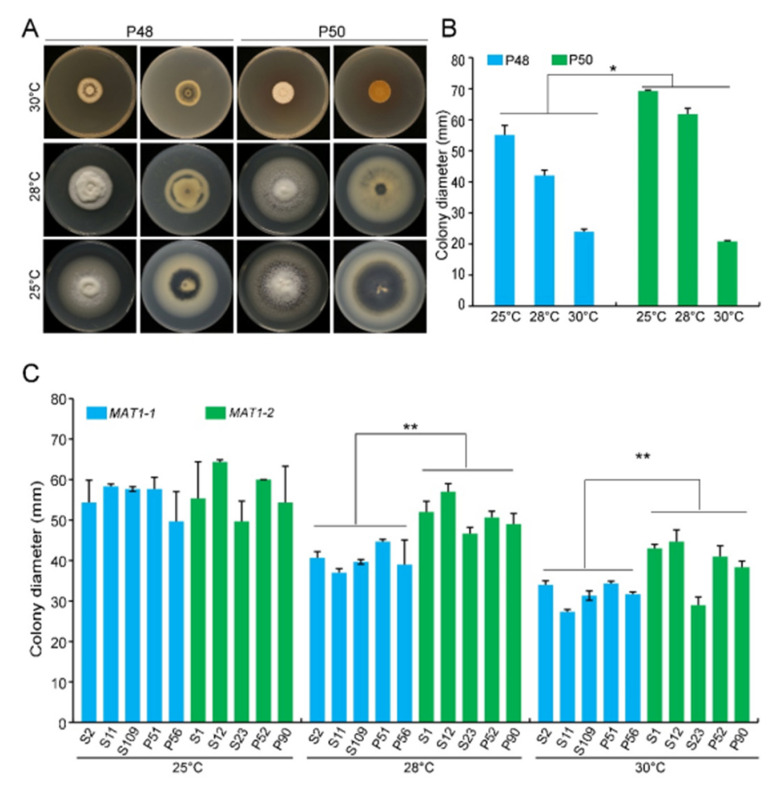
*MAT1-1* and *MAT1-2* strains of *Verticillium dahliae* grown under temperature stress conditions. (**A**) Growth phenotype of P48 and P50 on PDA medium at 25 °C, 28 °C, and 30 °C after culturing for 15 days. (**B**) Colony diameter of strains P48 and P50 on PDA medium at 25 °C, 28 °C, and 30 °C after culturing for 15 days. (**C**) Colony diameter of *MAT1-1* and *MAT1-2* strain collections on PDA medium at 25 °C, 28 °C, and 30 °C after culturing for 15 days. P and S indicate strains isolated from potato and sunflower, respectively. Asterisks * and ** indicate significant differences; *p* < 0.05 and *p* < 0.01, respectively, according to an unpaired Student’s *t*-test.

**Figure 3 ijms-22-07148-f003:**
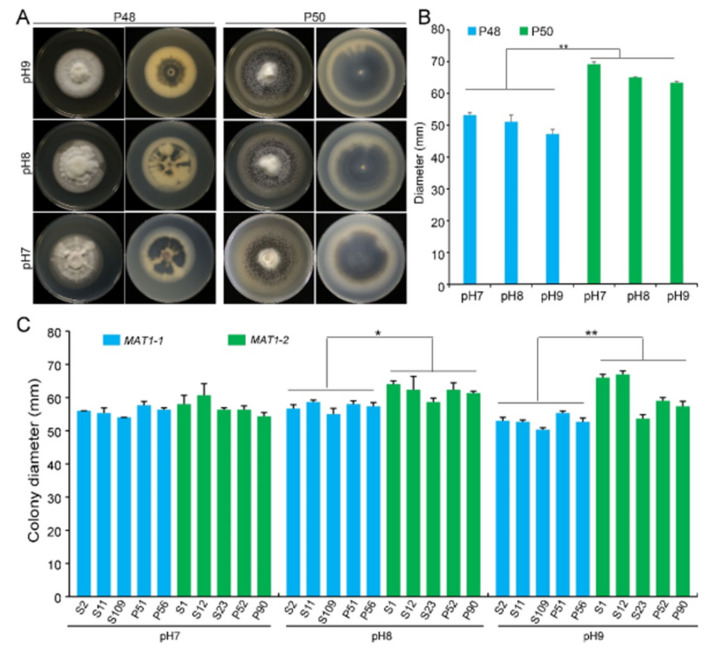
The influence of pH on the growth of *MAT1-1* and *MAT1-2* strains of *Verticillium dahliae*. (**A**) Growth phenotype of P48 and P50 at pH 7, pH 8, and pH 9 after culturing for 15 days. (**B**) Colony diameters of P48 and P50 at pH 7, pH 8, and pH 9 after culturing for 15 days. (**C**) Colony diameter of *MAT1-1* and *MAT1-2* strain populations at pH 7, pH 8, and pH 9 after culturing for 15 days. P and S indicate strains isolated from potato and sunflower, respectively. Asterisks * and ** indicate significant differences; *p* < 0.05 and *p* < 0.01, respectively, according to an unpaired Student’s *t*-test.

**Figure 4 ijms-22-07148-f004:**
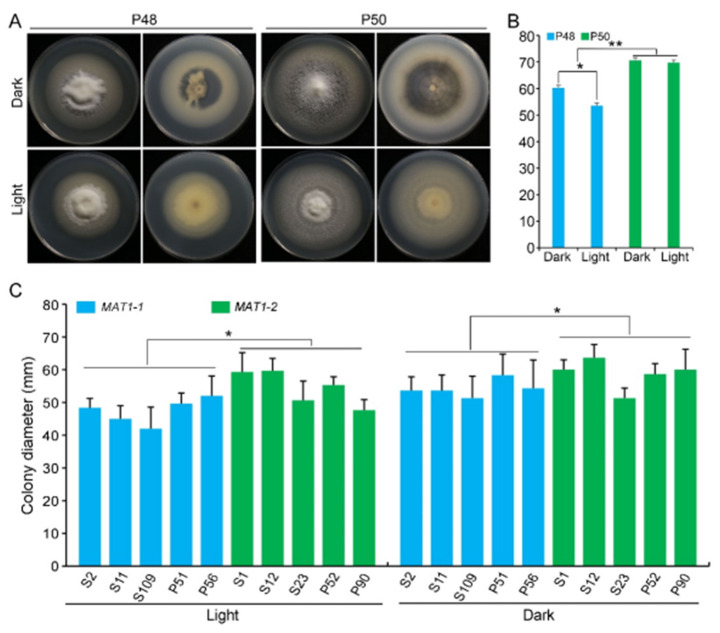
The effect of light and dark on *MAT1-1* and *MAT1-2* strains of *Verticillium dahliae*. (**A**) Growth phenotype of P48 and P50 under light and dark conditions after culturing for 15 days. (**B**) Colony diameters of P48 and P50 under light and dark conditions after culturing for 15 days. (**C**) Colony diameter of *MAT1-1* and *MAT1-2* strain populations under light and dark conditions after culturing for 15 days. P and S indicate strains isolated from potato and sunflower, respectively. Asterisks * and ** indicate significant differences; *p* < 0.05 and *p* < 0.01, respectively, according to an unpaired Student’s *t*-test.

**Figure 5 ijms-22-07148-f005:**
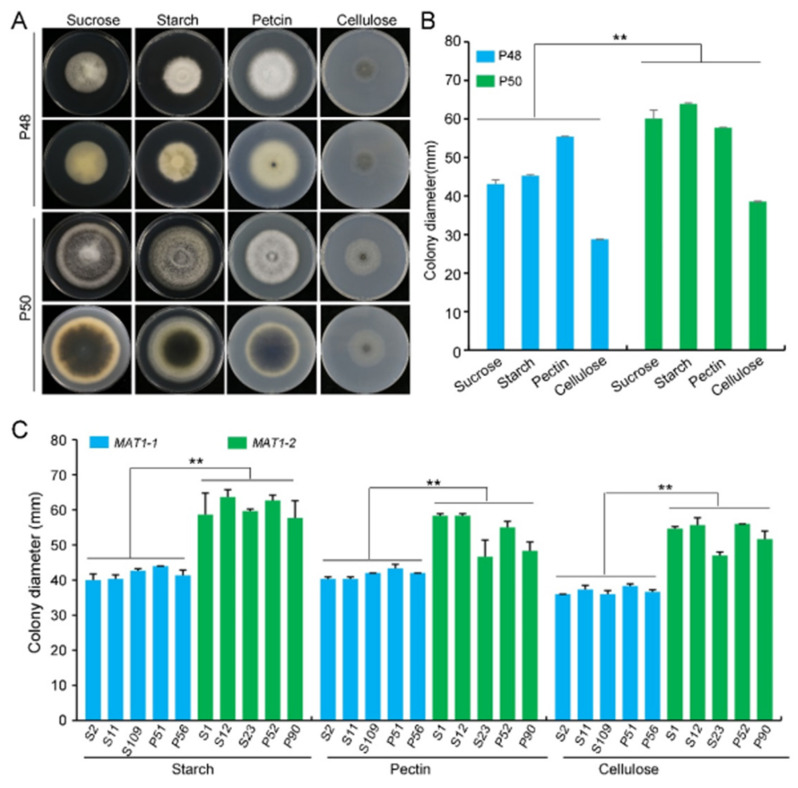
The influence of different carbon sources on the growth of *MAT1-1* and *MAT1-2* strains from *Verticillium dahliae*. (**A**) Growth of P48 and P50 on media types containing different carbon sources (sucrose, starch, pectin, cellulose). (**B**) Colony diameters of P48 and P50 on media types containing four carbon sources (sucrose, starch, pectin, cellulose) after culturing for 15 days. (**C**) Colony diameter of *MAT1-1* and *MAT1-2* populations on media types containing different carbon sources (sucrose, starch, pectin, cellulose) after culturing for 15 days. P and S indicate strains isolated from potato and sunflower, respectively. Asterisks ** indicate significant differences; *p* < 0.01, according to an unpaired Student’s *t*-test.

**Figure 6 ijms-22-07148-f006:**
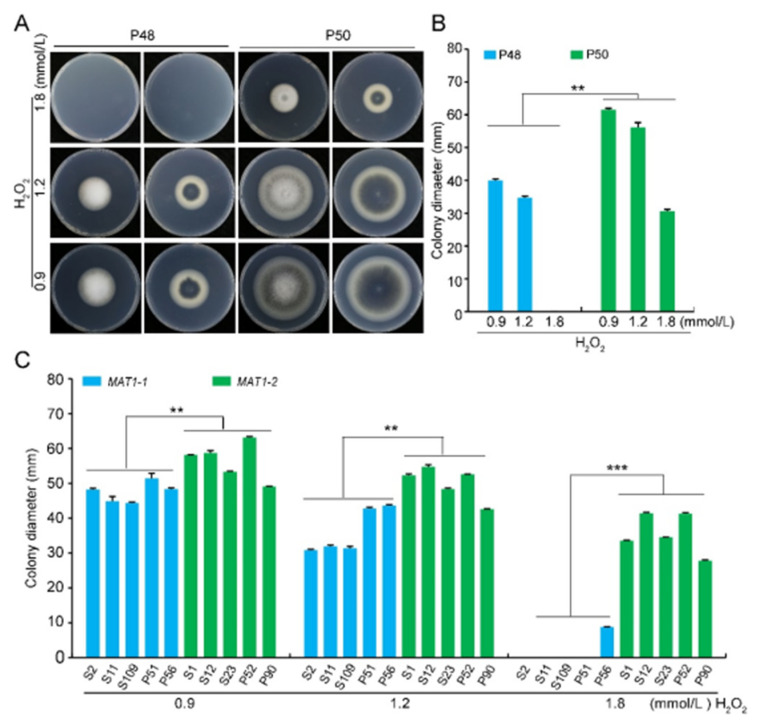
Growth of *MAT1-1* and *MAT1-2* strains of *Verticillium dahliae* in oxidative stress conditions. (**A**) Growth phenotype of P48 and P50 in response to serial concentrations of H_2_O_2_ (0.9, 1.2, 1.8 mmol/L). (**B**) Colony diameters of strains P48 and P50 after culturing for 15 days with different concentrations of H_2_O_2_ (0.9, 1.2, 1.8 mmol/L). (**C**) Colony diameters of *MAT1-1* and *MAT1-2* populations in response to serial concentrations of H_2_O_2_ (0.9, 1.2, 1.8 mmol/L) at 15 days of growth. P and S indicate strains isolated from potato and sunflower, respectively. Asterisks ** and *** indicate significant differences; *p* < 0.01 and *p* < 0.001, respectively, according to an unpaired Student’s *t*-test.

**Figure 7 ijms-22-07148-f007:**
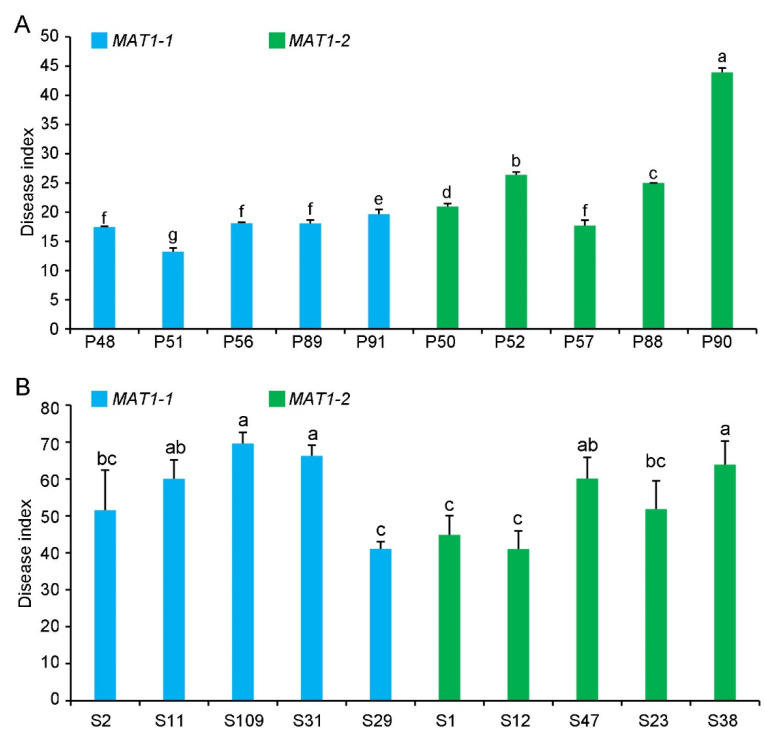
Pathogenicity of *MAT1-1* and *MAT1-2* strain populations of *Verticillium dahliae* on potato and sunflower. (**A**) Disease index of *MAT1-1* and *MAT1-2* strain populations isolated from potato (P) (*MAT1-1* strains: P48, P51, P56, P89, P91; *MAT1-2* strains: P50, P52, P57, P88, and P90) three weeks after inoculating potato (He 15). (**B**) Disease index of *MAT1-1* and *MAT1-2* strains isolated from sunflower (LD 5009) (S) (*MAT1-1* strains: S2, S11, S109, S31, and S29; *MAT1-2* strains: S1, S12, S47, S23, and S38) three weeks after the inoculation of sunflower. Columns with different letters represent a statistical significance of *p* < 0.05 according to Duncan’s new multiple range test.

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
