# Peer review of "Biological Characteristics of Verticillium dahliae MAT1-1 and MAT1-2 Strains"

_ijms, 2021, doi:10.3390/ijms22137148_

Round 1

Reviewer 1 Report

This is a well written manuscript, will clearly explained results, materials and methods, and significance.  Figure 1C was too difficult to read, the size needs to be increased.  

Line 335: The MAT1-2 strains from potato were more virulent than the MAT1-1 strains from potato.  This is not correct in regards to strain P57, a MAT1-2 strain, which had less virulence and was more similar to MAT1-1 strains.  

Figure 7B:  The mean separation letters did not appear to be correct.  Strain S47 had an ab designation, even though numerically it was one of the lowest in terms of virulence.  Strain S29 had a c designation, even though it was one of the highest in terms of virulence.  Why?

Line 468:  move the (10.0 g/L) to immediately after cellulose.

References

Be more consistent in formatting, especially with regards to journal name.

2. Phytopathology should not be abbreviated

3. Spacing is off.

4. Plant Disease (no J. Plant Dis.).

5. Same comment as 4.

31.  Italics for fungal name.

32. Do not capitalize the first letter of every word.

36. Has two different citations under this one.

59. Italics for the fungal species name.

You have not included some cited references including Fromm, 2013; Inderbitzin and Subbarao, 2014; Usami et al., 2017.  

I did not see the citation for Depotter et al., 2016.

Many of the citations are out of order.

Author Response

Reviewer 1

Comments and Suggestions for Authors

This is a well written manuscript, will clearly explained results, materials and methods, and significance.

Figure 1C was too difficult to read, the size needs to be increased. 

Many thanks! We have enlarged Figure 1C in the revised manuscript as we can.

Line 335: The MAT1-2 strains from potato were more virulent than the MAT1-1 strains from potato.  This is not correct in regards to strain P57, a MAT1-2 strain, which had less virulence and was more similar to MAT1-1 strains. 

We have rewritten this sentence to eliminate inaccuracies.

Figure 7B:  The mean separation letters did not appear to be correct.  Strain S47 had an ab designation, even though numerically it was one of the lowest in terms of virulence.  Strain S29 had a c designation, even though it was one of the highest in terms of virulence.  Why?

Thank you for pointing this out! We regret the errors in our previous data in Figure 7B. We have used the correct data and recalculated the statistical significance of all data in this revision.

Line 468:  move the (10.0 g/L) to immediately after cellulose.

Done

References

Be more consistent in formatting, especially with regards to journal name.

Done

  1. Phytopathology should not be abbreviated

Thanks! Corrected.

  1. Spacing is off.

Thanks! Corrected.

  1. Plant Disease (no J. Plant Dis.).

Thanks! Corrected.

  1. Same comment as 4.

Thanks! Corrected.

  1. Italics for fungal name.

Thanks! Corrected.

  1. Do not capitalize the first letter of every word.

Thanks! Corrected.

  1. Has two different citations under this one.

Thanks! Corrected.

  1. Italics for the fungal species name.

Thanks! Corrected.

You have not included some cited references including From 2013; Inderbitzin and Subbarao, 2014; Usami et al., 2017. 

Thanks! Corresponding literature is supplemented in the revised manuscript.

I did not see the citation for Depotter et al., 2016.

Thanks! Deleted!

Many of the citations are out of order.

Thanks! We adjusted the order of all references in the revised manuscript.

Reviewer 2 Report

The Manuscript entitled "Biological Characteristics of Verticillium dahliae MAT1-1 and MAT1-2 Strains" submitted to International Journal of Molecular Sciences presents a great piece of work and it will be of great interest to the scientific community working on these lines. I highly recommend this manuscript for publication.

Author Response

Many thanks for the reviewer's comments.